# Gender Self-Perception and Psychological Distress in Healthcare Students during the COVID-19 Pandemic

**DOI:** 10.3390/ijerph182010918

**Published:** 2021-10-17

**Authors:** Beatriz Rodríguez-Roca, Ana Belén Subirón-Valera, Ángel Gasch-Gallén, Estela Calatayud, Isabel Gómez-Soria, Yolanda Marcén-Román

**Affiliations:** 1Faculty of Health Sciences, University of Zaragoza, 50009 Zaragoza, Spain; Brodriguez@unizar.es (B.R.-R.); subiron@unizar.es (A.B.S.-V.); angelgasch@unizar.es (Á.G.-G.); yomarcen@unizar.es (Y.M.-R.); 2Institute for Health Research Aragón, 50009 Zaragoza, Spain

**Keywords:** gender self-perception, COVID-19, university health students, gender stereotypes, physical activity

## Abstract

The aim of this study was to analyze university Health Sciences students’ self-perception regarding gender stereotypes, and to explore whether there was any association between gender stereotypes and clinical/socio-demographic variables. Methods: This cross-sectional study was conducted with a sample of 252 university students who completed a self-administrated online questionnaire (18.3% males, 81.7% females). We evaluated the self-perception of gender stereotypes as determined using the BSRI-12 questionnaire and explored the association of this measure with the impact of perceived stress measured using a modified scale (PSS-10-C) as well as anxiety and depression according to scores on the Goldberg scale (GADS). Results: According to the students’ self-perception of gender stereotypes, 24.9% self-perceived themselves as feminine, 20.1% as masculine, 24.9% as androgynous, and 30% as undifferentiated. The degree determines self-identification with gender stereotypes. Nursing and Occupational Therapy are studied mostly by women, 28.4% and 45%, respectively, while Physiotherapy is studied mainly by men (71.2%). Females indicated more anxiety (75.7%) and depression (81.7%) than males (52.9% and 67.3%, respectively). In contrast, males developed more stress (88.5%) than females (74.1%). Conclusions: University degree, anxiety, depression, and stress determined self-identification with gender stereotypes. The results of this study indicate that gender roles influence the possibility of developing mental disorders and should be taken into account in future studies.

## 1. Introduction

The SARS-CoV-2 virus causing COVID-19 has brought about a global, shared reaction from governments worldwide, restricting citizens’ mobility and social contacts. According to international healthcare recommendations, some of these measures are still in force to a greater or lesser extent, and their main objective is to stop the spread of the virus [1]. Notwithstanding, the numbers of infected people worldwide have been very large [2], and subjective fear of contagion has been considerable [3].

The lockdown of cities or the whole country, imposing curfews, and the closure of education institutions were some of the preventive strategies implemented by the Spanish government to control the spread of COVID-19. The daily living situation has changed drastically, as have individual and family dynamics, with restricted travel, leisure activities and religious gatherings. The pandemic and its contingency measures have meant that other health-related activities have been left to one side. It is well-known that family stress is a major source of mental-health-related stress in adolescents [4]. Teenagers’ and young adults’ self-perception of isolation was greater in relation to COVID-19 and its effects on education conditions [5].

Several studies have already described the indirect impact of the COVID-19 outbreak on higher education. University students have experienced a high incidence of emotional disorders [6], with anxiety, depressive symptoms, suicidal ideation, and sleeping difficulties being common among these students [7]. Research has confirmed that adolescent students’ self-concept and scarce academic self-efficacy are an important source of stress and mental health disorders [8]. For this reason, it is important that studies about the pandemic consider how stereotypes and gender roles determine inequalities in the population’s health. Males and females have different health patterns, and these gender differences pose a challenge for the public health field [9], while the gender roles that have appeared for decades are related to inequities deriving from the imposition of sex-gender structure [10].

In order to measure gender stereotypes, several studies have applied the Bem Sex Role Inventory (BSRI) [11] to understand how self-identification with instrumental and expressive stereotypes impacts health outcomes [12]. Bem was the first to conceptualize gender roles as something other than exclusively masculine and feminine. According to Bem, gender stereotypes can be classified as masculine, feminine, androgynous, or undifferentiated [13,14].

Adhering to conventional gender stereotypes is important for analyzing the psychological well-being of females and males. One study found that the well-being of men and women whose self-concept included masculine-instrumental and feminine-expressive characteristics was better [11]. Contemporary studies have assessed that the best predictor of mental health was found to be masculinity rather than femininity and the results one study confirmed the androgyny model, highlighting the role of androgyny and masculinity as protective factors of mental health. The same study reported fewer gender-typed individuals and more flexible ways to adapt to gender roles in university students [15]. As previously mentioned, it is necessary to continue to gather evidence regarding sexual differences, and to avoid androcentric and dichotomic models because they can imply research biases and lead to health inequalities in research, teaching, and the healthcare practice [16].

The objective of this study was to understand university Health Sciences students’ self-identification with gender stereotypes. It also aimed to analyze whether there was any association between gender stereotypes and clinical/socio-demographic variables.

## 2. Materials and Methods

A cross-sectional descriptive study was performed on the identification of traits related to self-identification about gender stereotypes. An analysis of the association between the perceived stress caused by the COVID-19 pandemic and the characteristics of those students studying Health Sciences degrees was performed. 

### 2.1. Participants

Information was collected from the undergraduate students registered for the bachelor’s degrees of Nursing, Physiotherapy, and Occupational Therapy at the University of Zaragoza. For inclusion in the study participants had to answer all the Bem Sex Role Inventory questionnaire items (BSRI-12 items). Participants were excluded if they were not studying one of the degrees listed above.

Information was collected from students who were registered for the degrees taught at the Faculty of Health Sciences (Nursing, Physiotherapy, and Occupational Therapy) of the University of Zaragoza. We believe it is important to know how these students’ self-perceived gender may relate to stress, anxiety, and depression. These students will be future healthcare professionals and will attend people with health problems. It could be a priority to make the results visible and show them to academic leaders so they can adopt strategies that would minimize the risk of psychological stress to support students.

A sample size of 252 undergraduate students was estimated by taking a 4.65% error margin for the proportions for the frequency of results on these scales, as calculated by the EPIDAT 4.2 program. As a maximum 5% margin error is recommended, we believe that obtaining these data would mean that our study population size was sufficiently large.

### 2.2. Ethical Considerations

This study was performed according to the Declaration of Helsinki. Data were confidentially processed in line with Spanish Organic Law March 2018 on Personal Data Protection (LOPD). Consent came from the Committee of Research Ethics of the Spanish Autonomous Community of Aragón (CEICA) before the study began (Ref: C.P.-C.I. PI21/004).

### 2.3. Data Collection

An online questionnaire on socio-demographic variables was designed ad hoc and was later diffused via a web link to different student media in January 2021. Participation in this study was completely anonymous and voluntary. This study included the following socio-demographic variables: gender (male/female), age, degree (Nursing, Physiotherapy, Occupational Therapy), academic year (first, second, third, fourth), occupational status (working fulltime, working part-time, doing unpaid work, unpaid voluntary work), economic situation, place of residence (urban/rural), and birth country (Spain/elsewhere).

### 2.4. Data Analysis

In order to analyze the application and results of the self-identification instrument with gender stereotypes, an exploratory factor analysis of principal components and Varimax rotation with Kaiser correction of the BSRI-12 items was performed. Reliability was analyzed using Cronbach’s test and the application conditions were according to population size (KMO and Bartlett’s test of sphericity). The main factors that resulted with eigenvalues higher than 1 were retained. To assign each item to a factor, their factor loadings had to be over 0.5. The variable was recoded from the median of each of the two factors to obtain the four categories proposed by Sandra Bem (feminine, masculine, androgynous, undifferentiated).

In order to determine students’ psychological status, a questionnaire on stress caused by COVID-19 [17] and the Goldberg abbreviated anxiety and depression scale (GADS) [18,19] were employed. To measure perceived stress caused by COVID-19, the modified PSS-10 version related to COVID-19 (PSS-10-C) was applied. It comprises 10 items. Each offers five response options: “never”, “almost never”, “occasionally”, “almost always”, “always”. They are classified from 0 to 4. Items 1, 2, 3, 6, 9, and 10 are directly scored from 0 to 4. Items 4, 5, 7, and 8 are scored from 4 to 0. The higher the score, the greater the perceived stress. A cutoff point of ≥25 is related to high perceived stress for COVID-19 [20]. The abbreviated GADS measures anxiety and depression in the general population by examining four basic psychiatric areas: depression, anxiety, social anxiety disorder, hypochondria. This instrument has been previously validated and combines demonstrated applicability qualities. Cutoff points are equal to or exceed 4 for the anxiety score and are equal to or exceed 2 for depression. Sensitivity (83.0%) and specificity (81.8%) give a 95% positive predictive value [19].

The IPAQ questionnaire was used to analyze physical activity. It is a self-administered questionnaire consisting of six items on how many days a week light, moderate, and intense physical activity are practiced, and for how many minutes [21].

### 2.5. Statistical Analysis

A descriptive analysis was performed with frequency and percentages for the qualitative variables, and with mean and standard deviation for the quantitative variables. To compare the quantitative variables, the Kruskal–Wallis test was used. A bivariate study was performed to determine the dependence between two categorical variables by applying the statistical χ2 test. Logistic regression models were built to study the relation between the risk of depression and the different study variables. These models included variables showing a statistical significance of *p* < 0.2 in the bivariate study [22].

The significance level of all the analyses was set at *p* ≤ 0.05. Regardless of the encountered statistical significance, all variables were studied as a whole because all the clinical information that they offer needed to be assessed in a study such as the present one [23]. The quantitative data were analyzed with SPSS, version 26 (IBM, Corp., Chicago, IL, USA).

## 3. Results

In total, 252 Health Sciences students participated in this study 1 year after COVID-19 broke out. The mean age of participants was 21.02 years (SD 5.20) and 81.7% were female. The majority were born in Spain (92.1%). The following data were obtained: 74.6% lived in an urban area, 72.6% had never smoked, 84.5% had no chronic disease, 78.2% depended on family income, 8.8% did unpaid work, 89.7% did not have COVID-19, and 80.2% reported having no family relatives with COVID-19. Due to the COVID-19 pandemic, 71.4% had anxiety, 81% depression, and 13.1% stress. Table 1 and Table 2 show the characteristics of the general sample.

The factor analysis of the BSRI-12 instrument showed very good reliability in the study population, with a Cronbach’s alpha of 0.848 and good application conditions according to population size (KMO of 0.832 and Bartlett’s test of sphericity with *p* < 0.001). Two main factors appeared with eigenvalues higher than 1. The items included in each factor are shown in Table 3. The items of the first factor obtained scores between 0.680 and 0.816 and explained 38.20% of total variance, with Cronbach’s alpha of 0.872. Those of the second factor obtained scores between 0.507 and 0.840 and explained 19.19% of total variance, with Cronbach’s alpha of 0.820.

After dividing the total sample of students into females (*n* = 209) and males (*n* = 43), according to the self-perception of gender stereotypes, 24.9% self-perceived themselves as feminine, 20.1% as masculine, 24.9% as androgynous, and 30% as undifferentiated. Their degree determined their self-identification with gender stereotypes. Nursing and Occupational therapy were studied mostly by females (28.4% and 45% respectively), while Physiotherapy was studied mainly by males (71.2%) (Table 4). Females indicated more anxiety (75.7%) and depression (81.7%) than males (52.9% and 67.3%, respectively). In contrast, males indicated more stress (88.5%) (Table 5).

Gender role determined health outcomes, specifically the IPAQ, in such a way that the performance of light physical activity was statistically significantly different according to the self-perceived gender stereotypes, and the male specifically performed more light activity (*p* = 0.021) (Table 6).

## 4. Discussion

This study focused on understanding university Health Sciences students´ self-perception of gender stereotypes and the association of this perception with clinical and socio-demographic variables.

In our study, the highest proportion of the study sample was characterized in the gender role “undifferentiated”, where their main characteristic was their assertiveness. Among females, 30.2% self-identified as being undifferentiated, 22.3% as androgynous, 28.4% as feminine, and 19.1% as masculine. The study by Donelly el al. [24] revealed that the femininity levels for females have considerably lowered in recent years. In males, we found that students’ gender role was characterized as follows: 36.5% were androgynous, 30.8% undifferentiated, 25% masculine, and 7.7% feminine. One study by Szpitalak et al. [25] indicates that both men and women have masculine and feminine attributes, which means that gender roles are not inherent to sex.

In line with our results, one study in 2010 with a sample of 815 adolescents (M = 15.65 years, SD = 1.42) identified that 34.4% of young men and women did not conform to traditional gender stereotypes but defined themselves as androgynous. The subjects included in this category, along with those who were self-perceived with predominantly masculine characteristics, stated that they performed more sexual activity and were more erotophylic [26].

In this study, we found a significant association between gender role and what university degree students were studying. Students who perceived themselves more strongly as feminine, and whose associated characteristic was expressiveness, tended to study Nursing and Occupational Therapy degrees to a greater extent. This could be due to these Health Sciences degrees being more oriented toward care. This lies in contrast to the Physiotherapy degree, for which the predominant gender role was masculine, which could be due to physiotherapists applying more specific techniques to treat patients, and the degree being characterized as a more instrumental one. A study by Mesquita et al. [27] indicated that 13.1% of students selected social and health-related studies, 71% of whom were female. Different authors report that women are possibly more predisposed to professions that involve caring for other people [27,28].

Our study found an association between gender and mental health, where 75.7% of females indicated anxiety in comparison to 52.9% of males, and 81.7% of females scoring positively for depression versus 67.3% for males. Conversely, no significant data were obtained to explain the relation of gender roles with stress, anxiety, or depression. Similar to our study, a study performed in 2009 in Spain [29] with a sample of 337 people aged 17–74 years (M = 32.2 years, SD = 12.2), with different socio-demographic and gender-differentiated characteristics, found that women obtained higher mean scores for somatic symptoms, anxiety, and insomnia than men. The same study observed associations between gender roles and mental health (anxiety, insomnia, depression). The authors of that work also used the BSRI instrument and found that the men classified as undifferentiated presented fewer somatic symptoms compared to the subjects who classified as feminine or androgynous, whereas the women classified as undifferentiated indicated more somatic symptoms, anxiety, and insomnia than the androgynous subjects. Another study [30] revealed that subjects with more traits associated with masculinity were associated with being at a lower risk of depression.

The data that we collected indicate that those who self-identified as more feminine were 1.02-fold more likely to suffer stress. For depression, we observed that the androgynous subjects were 1.93-fold less likely to suffer depression, suggesting this gender role could act as a protective factor.

As pointed out by Lara in a study from 1991 [31], androgynous people are characterized by behavioral flexibility and emotional-personal adaptation, and this study also showed that undifferentiated subjects were more prone to depression and anxiety. Similarly, a study by Lin et al. [30] which was performed with students pointed out that the university students with fewer traits of masculinity (regardless of gender) were very vulnerable to depression during the COVID-19 outbreak. This revealed that gender role influenced the possibility of developing mental disorders.

Some studies have indicated a relation between mental problems and physical activity [32,33]. The COVID-19 pandemic has been reported to bring about a change in the general population’s habits, and physical activity diminished mainly due to restrictions and confinement [34,35,36]. It is worth pointing out that physical activity levels were well below the 150–300 min/week recommended by the WHO for moderate to strenuous physical activity [37]. Our study found a significant association between gender role and light physical activity. A systematic review by Mammen et al. [38] reveals that physical activity is negatively associated with the risk of suffering depression later. Most reviewed studies included a high-quality methodology and postulated that physical activity could prevent depression in the future. The study by Lin et al. [30] indicated that a higher risk of developing depression was associated with a lower physical activity level during the present pandemic. Finally, the university students who reported fewer masculine traits (regardless of gender) were more susceptible to depression during the COVID-19 outbreak.

Our results show that it is possible to describe different gender roles and stereotypes in the Health Sciences students at the University of Zaragoza. Differences were observed in students’ perceptions according to the degrees they were studying and in how these perceptions were related to stress, depression, and anxiety.

Moving forward, it is necessary to consider factors, such as the creation of new scenarios and policies that promote the departure from traditional gender roles. It is also necessary to design and set up intervention strategies with an integrated approach, as well as major structural education reforms, with a wide range of policies and programs in both public and private domains.

When designing institutional policies such as university syllabi to improve well-being, specific strategies could be included. Some Nursing studies deal with the importance of designing educational Nursing curricula that reflect on gender stereotypes [39,40].

Another advantage of the use of questionnaires regarding the self-perception of gender roles is that they could help teachers to know which of their students might more easily develop psychological disorders like stress and anxiety, and this could be considered when designing objectives and activities to acquire competences.

The strategies put into practice by higher education result from sporadic measures that which merely patch up specific problems and ignore very important matters that affect what is structural, institutional, and even social. Undoubtedly, a new and more structured approach is needed.

## 5. Conclusions

Health Sciences students’ self-perception regarding gender indicates a distance from traditional gender stereotypes, since a higher frequency of participants in the undifferentiated category was observed. We also observed current trends about the students involved in these programs of study. For example, a greater proportion of women showed anxiety and depression, and women chose to study physiotherapy at lower rates than men. According to our results, there was an association between instrumental traits, self-identification, and health outcomes, benefiting those who practiced light physical activity and protecting them from mental health problems.

Our results suggest that involvement in expressive gender stereotypes could be a kind of reflection of educational and health inequalities derived from gender impositions on our students. Those who self-perceived near to traditional masculine stereotypes seemed to have more opportunities to perform physical activity, select specific studies, and have more protection against mental disorders.

### Implications for Research and Interventions

One of the study limitations is that it was conducted during pandemic times. This means that its data must be cautiously shown because the characteristics offered by students were collected during a specific time period. For this research line to be more comprehensive, comparative studies with students learning different degrees and from distinct cultures worldwide must be carried out.

University educational policies should be addressed to identify gender stereotype adherence amongst students, and should offer more support for individuals and groups presenting more vulnerable self-identification traits.

## Figures and Tables

**Table 1 ijerph-18-10918-t001:** Socio-demographic characteristics of the study sample.

Variables	Total (*n* = 252)
	Mean (SD)
Age	21.02 (5.20)
	Frequencies (%)
Gender	
Male	46 (18.3%)
Female	206 (81.7%)
Birth Country	
Spain	232 (92.1%)
Elsewhere	20 (7.9%)
Place of Residence	
Rural	64 (25.4%)
Urban	188 (74.6%)
Economic situation	
Grant-holder	34 (13.5%)
Depend on family income	197 (78.2%)
Independent	17 (6.7%)
Others	4 (1.6%)
Smoking	
Ex-smoker	15 (6%)
Occasional smoker	25 (9.9%)
No	183 (72.6%)
Yes	29 (11.5%)
Chronic disease	
No	213 (84.5%)
Yes	39 (15.5%)
First year	
Nursing	22 (8.7%)
Physiotherapy	22 (8.7%)
Occupational therapy	51 (20.2%)
Second year	
Nursing	11 (4.4%)
Physiotherapy	31 (12.3%)
Occupational therapy	12 (4.8%)
Third year	
Nursing	23 (9.1%)
Physiotherapy	13 (5.2%)
Occupational therapy	8 (3.2%)
Fourth year	
Nursing	19 (7.5%)
Physiotherapy	15 (6%)
Occupational therapy	23 (9.1%)
Missing	2 (0.8%)
Occupational status	
Do not work	171 (67.9%)
Others	12 (4.8%)
Working fulltime	17 (6.7%)
Working part-time	30 (11.9%)
Unpaid work	22 (8.8%)

**Table 2 ijerph-18-10918-t002:** Well-being/health characteristics.

Variables	Total (*n* = 252)
	Frequencies (%)
Stress (PSS-10-C)	
No	219 (86.9%)
Yes	33 (13.1%)
Anxiety (GADS)	
No	68 (27%)
Yes	180 (71.4%)
Missing	4 (1.6%)
Depression (GADS)	
No	47 (18.7%)
Yes	204 (81%)
Missing	1 (0.4)
Confined by COVID-19	
No	140 (55.6%)
Yes	112 (44.4%)
Had COVID-19	
No	226 (89.7%)
Yes	26 (10.3%)
Relative had COVID-19	
No	202 (80.2%)
Yes	50 (19.8%)

**Table 3 ijerph-18-10918-t003:** Gender role characteristics (BSRI) in undergraduate Health Sciences students.

	Femenine Factor	Masculine Factor
BSRI_GENTLE	0.725	
BSRI_COMPRESSIVE	0.794	
BSRI_HAVE LEADERSHIP ABILITIES		0.840
BSRI_ACT AS A LEADER		0.784
BSRI_DOMINANT	0.798	
BSRI_TENDER	0.796	
BSRI_WARM	0.816	
BSRI_AFFECTIONATE		0.639
BSRI_STRONG PERSONALITY		0.507
BSRI_DEFENDER OF OWN BELIEFS	0.680	
BSRI_SENSITIVE TO OTHERS´ NEEDS		0.554
BSRI_MAKES DECISIONS EASILY		0.871

**Table 4 ijerph-18-10918-t004:** Self-perception of gender stereotypes (distribution in instrumental/expressive-type categories) and characteristics per gender in Health Sciences students.

	Females (*n* = 209)	Males (*n* = 43)	Overall (*n* = 252)	*p*
	Frequencies (%)	Frequencies (%)	Frequencies (%)	
*Self-perception of gender stereotypes*				
Feminine	59 (28.4)	3 (7.7)	62 (24.9)	0.039 ^a^
Masculine	40 (19.1)	11 (25)	51 (20.1)	
Androgynous	47 (22.3)	16 (36.5)	63 (24.9)	
Undifferentiated	63 (30.2)	13 (30.8)	76 (30)	
*Socio-demographic, educational and health variables*				
	Mean (SD)	Mean (SD)	Mean (SD)	
Age (years)	20.63 (4.57)	22.71 (7.50)	20.94 (5.17)	0.120 ^b^
	Frequencies (%)	Frequencies (%)	Frequencies (%)	
Birth Country				
Spain	190 (91%)	41 (96.2%)	231 (91.6%)	0.111 ^b^
Others	19 (9%)	2 (3.8%)	21 (8.4%)	
Place of Residence				
Rural	59 (28.4%)	7 (17.3%)	66 (26.8%)	0.119 ^b^
Urban	150 (71.6%)	36 (82.7%)	186 (73.2%)	
Economic situation				
Grant-holder	29 (14%)	4 (9.6%)	33 (13.3%)	0.618 ^b^
Depend on family income	167 (80.2%)	32 (75%)	199 (79.5%)	
Independent	9 (4.7%)	6 (13.5%)	15 (6%)	
Others	4 (1.1%)	1 (1.9%)	5 (1.2%)	
Degree				
Nursing	59 (28.4%)	8 (19.2%)	67 (26.8%)	0.000 ^b^
Physiotherapist	53 (25.5%)	30 (71.2%)	83 (32.5%)	
Occupational therapy	94 (45%)	5 (9.6%)	99 (39.85%)	
Missing	3 (0.9%)	-		

Chi-square test ^a^; Kruskal–Wallis test ^b^.

**Table 5 ijerph-18-10918-t005:** Characteristics per gender in students’ well-being and health.

	Females (*n* = 209)	Males (*n* = 43)	Overall (*n* = 252)	*p*
	Frequencies (%)	Frequencies (%)	Frequencies (%)	
Confined by COVID-19				
No	122 (58.3%)	17 (40.4%)	139 (55.4%)	0.052
Yes	87 (41.7%)	26 (59.6%)	113 (44.6%)	
Has COVID-19				
No	184 (88.1%)	42 (98.1%)	226 (89.8%)	0.176
Yes	25 (11.9%)	1 (1.9%)	26 (0.2%)	
Relative with COVID-19				
No	168 (80.6%)	35 (80.8%)	203 (80.7%)	0.923
Yes	41 (19.4%)	8 (19.2%)	49 (19.3%)	
Anxiety (GADS)				
No	51 (24.3%)	20 (47.1%)	71 (28%)	0.003
Yes	158 (75.7%)	23 (52.9%)	181 (72%)	
Depression (GADS)				
No	38 (18.3%)	14 (32.7%)	52 (20.8%)	0.023
Yes	171 (81.7%)	29 (67.3%)	200 (79.2%)	
Stress (PSS-10-C)				
No	54 (25.9%)	5 (11.5%)	59 (24.1%)	0.010
Yes	155 (74.1%)	38 (88.5%)	193 (75.9%)	

Chi-square test.

**Table 6 ijerph-18-10918-t006:** Relation between gender role and health variables.

	Feminine Median (RIQ)	Masculine Median (RIQ)	Androgynous Median (RIQ)	Undifferentiated Median (RIQ)	*p*
Light physical activity (IPAQ)	1	2.50	2	2	0.021
Moderate physical activity (IPAQ)	1	1	1	2	0.411
Intense physical activity (IPAQ)	7	7	7	7	0.665
Anxiety (GADS)	18	18.50	18	17	0.585
Depression (GADS)	6	4	6	6	0.101
Stress (PSS-10-C)	4	4	3	4	0.504

*p* = Kruskal–Wallis test.

## Data Availability

The data presented in this study are available on request from the corresponding author. The data are not publicly available due to specific requirements from the clinical research ethics committee that reviewed and approved this investigation.

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
