# Peer review of "Gender Self-Perception and Psychological Distress in Healthcare Students during the COVID-19 Pandemic"

_ijerph, 2021, doi:10.3390/ijerph182010918_

Round 1

Reviewer 1 Report

Review IJERPH

This study aimed at analyzing “the university Health Sciences students’ self-perception with gender stereotypes, clinical and socio-demographic variables.” We believe the authors do not have at this point the final version of their manuscript for publication. Unfortunately, we suggest a rejection, but we hope they improve it and re-submit.

Please, find below some points that led us to this decision.

First, we strongly advise the authors to re-read their manuscript as there are multiple English problems. These problems hinder the readability of their work, reducing their potential readers’ interest in their research.

Second, it regards the authors’ objective.

  • While in the Abstract they indicate one objective, in their Discussion section, the objective shifts to: “knowing the self-perception of university students of Health Sciences with gender stereotypes, in addition to analyzing the association between gender stereotypes and clinical/socio-demographic variables, identified that each degree of this faculty of Health Sciences self-identifies with different gender roles and that these influence the perception of stress, anxiety and depression.” This shift confuses the reader, changes their expectations, and most importantly, turns the objective unclear.

  • Departing from the first suggested objective, it is also unclear its relevance. We observe that the authors tried in the Introduction to define their contribution. Nevertheless, they were unable to prove their point, failing to provide clear reasons to “consider how stereotypes and gender roles determine the inequalities they found in this population’s health.” Probably, the authors will need to reformulate this section to clarify their contributions. We suggest re-writing this section from scratch.

Third, in terms of Methods, it seems that they are mixed, unclear, or not considering enough of the hypothesis needed for using the methods. We suggest the following readings:

  • Exploratory Factor Analysis (EFA): http://www.ffzg.unizg.hr/psihologija/phm/nastava/Book_Exploratory%20Factor%20Analysis.PDF and https://doi.org/10.1207/s15327752jpa6803_5. The first is a book that probably will discuss all possible problems. In contrast, the second is a paper where the authors may find the critical topics on EFA. For instance, the authors should provide a clear explanation on why they selected Varimax rotation (these researches might be helpful: https://doi.org/10.7275/hb2g-m060 or https://doi.org/10.1080/00273170903504810) instead of others. Furthermore, the KMO criterion and Bartlet’s sphericity test are related to EFA. Thus we suggest the authors discuss these results always near-by EFA.

  • Kruskal-Wallis and ANOVA: We observed that the authors were correctly worried about the normality assumption to choose which analysis they would apply. Nevertheless, the readers must know which test(s) they used in addition to the results for this hypothesis. If the authors do not feel this is cardinal to their analysis, they can provide this information with supplementary material.

  • Logistic regression: Why did the authors use the p < 0.2 cut-offs? Are they be the best criteria for this research? Which are the pros and cons of using this criterion for the study? Using this cut-off is better for defining a model than evaluating the possible goodness of fit criteria for the estimated models? All these answers must be answered in the text.

Fourth, most of the authors’ results are not discussed in the Discussion section. We suggest unifying these sections. Remember that the readers tend to lose most of the probable impactful findings when both sections are detached. In turn, this may reduce the impact of the authors’ research.

For instance, in lines 174-6, the authors indicated, “The following data were obtained: 74.6% lived in an urban area, 72.6% had never smoked, 84.5% had no chronic disease, 78.2% depended on family income and 8.8% did unpaid work, 89.7% did not have COVID-19 and 80.2% reported no family relative with COVID-19.” However, why is this information relevant? Does this indicate that the sample follows represents the Spanish population in Higher Education? Do they provide any critical trends?

Another example, in lines 271-3, the authors suggest, “The data that we collected indicate that those self-identified as more feminine are 1.4-fold more likely to suffer stress. For depression, we observe that the androgynous subjects are 3.3-fold less likely to suffer depression, which could act as a protector factor.” These results come from the logistic regression, although the authors did not provide the regression results besides the odds ratio (again, you might use supplementary material).

Finally, the authors abruptly conclude their research with one paragraph (three if we consider the implications for research and interventions). Just as the Introduction, we suggest re-writing this section, as their results may have other impacts, such as governmental (public policies?) or institutional (private policies?).

We hope these points do not discourage the authors, and we are confident that with more time and effort by their side, their research will have the impact they expect.

Reviewer 2 Report

1) The author should include the justification of choosing students among Bachelor of Nursing, Physiotherapy and Occupational Therapy and excluded student who were not studying any of the degree.

2) The author should stated how they determine the sample size.

3) Missing value of the analysis- The author should justify the missing values or else the author should do remedy on the missing value and run again the analysis.

Author Response

Thank you very much for your comments, we have included all your comments in the manuscript.

 All the changes in the manuscript will be in blue.

Comment 1: The author should include the justification of choosing students among Bachelor of Nursing, Physiotherapy and Occupational Therapy and excluded student who were not studying any of the degree.

Response 1: Thank you for the input. The justification of the choice of these students has been included in the text.

Comment 2: The author should stated how they determine the sample size.

Response 2: This information can be found in the texts on lines 123-127.

Comment 3: Missing value of the analysis- The author should justify the missing values or else the author should do remedy on the missing value and run again the analysis.

Response 3: We have added a table with the logistic regression in a supplementary material.

Round 2

Reviewer 1 Report

Dear authors,

thanks for considering our suggestions.  Once again I would suggest an improvement in Conclusion session, but that is up to you.

Best regards,

Author Response

Thank you for your suggestions, we will proceed to improve the conclusions.
Thank you very much for your contributions to the article.
